# TET2 Mutation May Be More Valuable in Predicting Thrombosis in ET Patients Compared to PV Patients: A Preliminary Report

**DOI:** 10.3390/jcm11226615

**Published:** 2022-11-08

**Authors:** Ziqing Wang, Weiyi Liu, Dehao Wang, Erpeng Yang, Yujin Li, Yumeng Li, Yan Sun, Mingjing Wang, Yan Lv, Xiaomei Hu

**Affiliations:** 1Department of Hematology, Xiyuan Hospital, China Academy of Chinese Medical Sciences, Beijing 100091, China; 2Graduate School, Beijing University of Chinese Medicine, Beijing 100029, China; 3Graduate School, China Academy of Chinese Medical Sciences, Beijing 100700, China

**Keywords:** TET2 mutation, essential thrombocythemia, polycythaemia vera, thrombosis, risk factors, clinical characteristics

## Abstract

Thrombosis is a common complication of myeloproliferative neoplasm (MPN), and it is a major cause of disability and death. With the development of next-generation gene-sequencing technology, the relationship between non-driver mutations and thrombotic risk factors has also attracted considerable attention. To analyze the risk factors of thrombosis in patients with essential thrombocythemia (ET) and polycythemia vera (PV), we retrospectively analyzed the clinical data of 125 MPN patients (75 ET and 50 PV) and performed a multivariate analysis of the risk factors of thrombosis using a Cox proportional risk model. Among the 125 patients, 35 (28.0%) had thrombotic events, and the incidence of thrombotic events was 21.3% and 38.0% in ET and PV patients, respectively. In ET patients, the multivariate analysis showed that a TET2 mutation and history of remote thrombosis were independent risk factors for thrombosis in ET patients, with an HR of 4.1 (95% CI: 1.40–12.01; *p* = 0.01) for TET2 mutation and 6.89 (95% CI: 1.45–32.68; *p* = 0.015) for a history of remote thrombosis. In PV patients, the multivariate analysis presented the neutrophil-to-lymphocyte ratio (NLR) (HR: 4.77, 95% CI: 1.33–17.16; *p* = 0.017) and a history of remote thrombosis (HR: 1.67, 95% CI: 1.03–1.32; *p* = 0.014) as independent risk factors for thrombosis, with no significant change in the risk of thrombosis in patients with TET2 mutations. A further analysis of the clinical characteristics and coagulation occurring in ET patients with a TET2 mutation revealed that the values of age and D-dimer were significantly higher and antithrombin III was significantly lower in TET2-mutated ET patients compared to TET2-unmutated patients. In summary, TET2 mutation may be more valuable in predicting thrombosis in ET patients than in PV patients. ET patients with a TET2 mutation are older and present differences in coagulation compared to TET2-unmutated patients.

## 1. Introduction

Myeloproliferative neoplasm (MPN) is a group of malignant hematologic diseases that occur with the excessive clonal proliferation of one or more lineages of myeloid cells in hematopoietic stem or progenitor cells, with one or more peripheral blood cells increasing as the main clinical feature, often accompanied by hepatosplenomegaly, thrombosis, a bleeding tendency, and extramedullary hematopoiesis [1]. The World Health Organization (WHO) defines essential thrombocythemia (ET), polycythemia vera (PV) and primary myelofibrosis (PMF) as classical BCR/ABL-negative MPNs. Since 2005, some scholars first discovered that the mutation rate of the JAK2V617F gene in BCR/ABL-negative MPN was greater than 50% [2]; in the following decade, mutations occurring in MPLW515, exon 12 of JAK2, and exon 9 of CALR were successively identified and reported [3,4,5]. At present, JAK2, CALR, and MPL are known as driver genes of MPN and are closely associated with disease development and thrombotic events [6].

Thrombosis is a common complication of MPN, including arterial or venous thrombosis and microvascular thrombosis, and is a major cause of disability and death in MPN patients. Thrombotic events are reported to occur in more than 30% of MPN patients, and thrombosis-related mortality accounts for 35–70% of total mortality rates [7]. Recent studies have presented a strong link between driver genes and the occurrence of thrombotic events. JAK2 gene mutation leads to the persistent activation of JAK-STAT and downstream signaling pathways, which not only increases the number of blood cells present, but also activates the production of various inflammatory cytokines, which in turn damage the vascular endothelium and eventually lead to thrombosis [8]. The risk factors of thromboembolism in MPN patients differ in ET and PV patients. For ET, the International Prognostic Score for ET Thrombosis (IPSET-thrombosis) included patients aged >60 years, cardiovascular risk factors, previous history of thrombosis, and JAK2V617F mutation as risk factors in the score system [9]. In PV, studies have shown that age, cardiovascular risk factors, and previous thrombotic history are also strongly associated with thrombosis [10,11], and no association was observed between JAK2V617F mutation and thrombosis risk, but some studies have shown that patients with a JAK2V617F-variant allele frequency (VAF) >50% are at a higher risk of venous thrombosis [12].

In recent years, with the development of research and the development of next-generation sequencing (NGS) technology, some studies have determined other non-driver mutations in MPN, such as TET2, ASXL1, DNMT3A, TP53, EZH2, IDH1, and IDH2. An increasing number of studies have shown that non-driver gene mutations are closely associated with thrombosis, fibrosis progression, and survival in MPN patients [13,14,15]. In an age-matched case–control study of PV patients, investigators observed that the presence of ≥1 TET2 or ASXL1 or DNMT3A mutations significantly increased the risk of thrombosis, and that PV patients with a TET2 mutation had a significantly higher risk of thrombosis [13]. In addition, studies have determined that ET patients with an ASXL1 mutation have a significantly higher incidence of thrombotic events [16]. In this study, we analyzed the risk factors for thrombotic events in 125 patients with PV and ET, applied next-generation sequencing technology to detect 31 MPN-related gene mutations, and observed that the TET2 mutation is an independent risk factor for thrombosis in ET patients, but no such association was observed in PV patients. Then, we further analyzed the clinical characteristics and coagulation indexes of ET patients with TET2 mutations, which are reported in the present study.

## 2. Methods

### 2.1. Patients

This was a single-center, retrospective study in which we analyzed patients with ET and PV attending the Xiyuan Hospital of China Academy of Chinese Medical Science between January 2018 and March 2022 and reassessed the diagnoses according to the WHO (2016) diagnostic criteria for ET and PV [17], retrospectively recording the baseline clinical and laboratory data. The patient selection process is illustrated in Figure 1. Finally, 125 patients with MPN who met the diagnosis and had complete clinical data were included, including 75 cases of ET and 50 cases of PV. DNA samples for NGS and peripheral blood samples for coagulation testing were obtained from the peripheral blood specimens collected at the time of diagnosis or within 3 months of diagnosis, but not upon treatment. This study was approved by the Ethics Committee of Xiyuan Hospital of China Academy of Chinese Medical Science according to the guidelines presented in the Declaration of Helsinki (2019XLA024-3). All the subjects signed the informed consent form.

### 2.2. Definition of Events

Thrombotic events included arterial and venous thrombosis. Arterial thrombosis included ischemic stroke, acute coronary syndrome, splenic thrombosis, and peripheral arterial thrombosis. Venous thromboembolism included deep vein thrombosis, pulmonary embolism and visceral vein thrombosis and peripheral vein thrombosis. Thrombotic events were confirmed by imaging and the imaging date was the time of the event. Splenomegaly was based on an abdominal ultrasound report. Thrombotic events that occurred within 2 years prior to the diagnosis of MPN were defined as MPN-related thrombotic events, and thrombotic events that occurred more than 2 years prior to the diagnosis were defined as history of remote thrombosis [18]. Thrombosis-free survival (TFS) was defined as the time interval from 2 years before diagnosis to the first case of thrombosis or last contact. Cardiovascular risk factors included smoking, hypertension, diabetes mellitus and hyperlipidemia.

### 2.3. Gene Testing

The mutation analysis of DNA obtained from the peripheral blood using NGS technology was performed. A total of 5 mL of venous blood was drawn in the early morning under fasting conditions, peripheral blood single-nuclei cells were extracted and whole genomic DNA specimens were prepared, and the samples with a qualified DNA concentration and purity determination were stored at −80 °C for backup. The DNA quality control samples were fragmented; the fragmented double-stranded DNA was end-repaired, PCR-amplified, and the products were purified; and the qualified Illumina sequencing platform standard libraries were obtained through detection. The successfully constructed genomic library was specifically hybridized using the liquid-phase microarray of the integrated genetic assay panel for hematological diseases to enrich the genomic region of the target area. The enriched DNA fragments were eluted down, the unbound DNA fragments were eluted and purified, the captured fragments were amplified by PCR, the captured libraries were subjected to Agilent 2100 and QPCR quality control, and the quality control-qualified samples were subjected to Illumina HiSeq2500 high-throughput sequencing. The filtered data were compared to the human reference genome HG19 using Burrows–Wheeler alignment and quality control of the corresponding indicators, and the output data were counted. Single-nucleotide variants (SNPs), small-fragment insertion–deletion variants (InDel), and mutation hotspots were detected using GATK to annotate and count variants in 31 genes associated with MPN, including JAK2, MPL, CALR, CSF3R, ASXL1, EZH2, TET2, IDH1, IDH2, SRSF2, SF3B1, CBL, DNMT3A, IKZF1, KIT, ABL, SETBP1, SH2B3, KRAS, NRAS, ETV6, GATA2, RUNX1, NF1 TP53, U2AF1, CBLC, CSF1R, EGFR, FAM46C, and SMC3.

### 2.4. Statistical Analysis

Continuous variables were described by the median and range values, and differences were assessed using the Mann–Whitney U test for between-group comparisons; categorical variables were described by the number of cases and percentages, and differences were assessed by χ^2^ or Fisher’s exact test for between-group comparisons. A Cox proportional hazards model was used for the multivariate analysis of risk factors for thrombosis. The Kaplan–Meier method was used for the survival analysis, and Kaplan–Meier survival curves were plotted and compared using the log-rank test. A difference of *p* < 0.05 was considered statistically significant. All analyses were performed using SPSS 23 (IBM, Armonk, NY, USA).

## 3. Results

### 3.1. Clinical Characteristics of 125 Patients with MPN

A total of 125 MPN patients, 67 (53.6%) males and 58 (46.4%) females, were evaluated, with a median age at diagnosis of 50 (18–78) years. The mutation rates of driver genes JAK2V617F, CALR, and MPL were 70.4%, 12.8%, and 0.8%, respectively. At least one non-driver mutation was detected in 36% of the patients, with the highest rate of mutations in TET2 (12%), followed by ASXL1 (9.6%). The distribution of mutated genes in ET and PV patients is presented in Figure 2A, and the number and frequency of mutations are presented in Figure 2B. Of the 125 MPN patients, 75 (60%) were ET and 50 (40%) were PV. Detailed clinical characteristics of patients with PV and ET are exhibited in Table 1. The proportion of male patients with PV and the proportion of patients with splenomegaly and JAK2V617F-mutation load were significantly higher than those in ET patients (*p* < 0.01).

### 3.2. Incidence of Thrombotic Events in MPN Patients

During a median follow-up of 62 months, the MPN-related thrombotic events occurred in 35 of 125 patients (28.0%), and the incidence of thrombotic events was 21.3% (16/75) in ET patients and 38.0% (19/50) in PV patients, respectively. Two patients presented with recurrent thrombosis, both with PV. Ischemic stroke (17.6%, 22/125) was the most common thrombotic event, followed by acute coronary syndrome (7.2%, 9/125), in addition to two cases of deep venous thrombosis of the lower limbs (1.6%), two cases of pulmonary embolism (1.6%), one case of splenic infarction (0.8%), and one case of fundic artery embolism (0.8%) (Figure 3). Eight patients had a history of remote thrombosis, including three cases of ET and five cases of PV.

### 3.3. Analysis of Risk Factors of MPN-Related Thrombotic Events

For 125 MPN patients, univariate analysis revealed significant associations between thrombosis and age at diagnosis (*p* < 0.001), age >60 years at diagnosis (*p* = 0.001), JAK2V617F mutation (*p* = 0.045), TET2 mutation (*p* = 0.004), WBC (*p* = 0.019), NLR (*p* = 0.002), cardiovascular risk factors (*p* = 0.039), and history of remote thrombosis (*p* = 0.001) (Table 2). In the multivariate analysis that included the above eight factors as covariates, age at diagnosis (HR: 1.03, 95% CI: 1.00–1.06; *p* = 0.046), TET2 mutation (HR: 2.84, 95% CI: 1.26–6.36; *p* = 0.011), NLR (HR: 1.18, 95% CI: 1.06–1.32; *p* = 0.003) and history of remote thrombosis (HR: 5.33, 95% CI: 1.93–14.70; *p* = 0.001) remained risk factors for thrombosis in MPN patients (Table 2).

In the subgroup analysis of ET and PV patients, a univariate analysis was performed for the parameters listed in Table 2, and factors with differences were used as covariates for further multivariate analyses. In ET patients, the multivariate analysis showed that TET2 mutation and a history of remote thrombosis were independent risk factors for thrombosis in ET patients, with an HR of 4.1 (95% CI: 1.40–12.01; *p* = 0.01) for TET2 mutation and 6.89 (95% CI: 1.45–32.68; *p* = 0.015) for history of remote thrombosis (Table 3). In PV patients, the multivariate analysis assessed the neutrophil-to-lymphocyte ratio (NLR) (HR: 4.77, 95% CI: 1.33–17.16; *p* = 0.017) and a history of remote thrombosis (HR: 1.67, 95% CI: 1.03–1.32; *p* = 0.014) as independent risk factors for thrombosis; however, no association of TET2 mutation with thrombosis was observed in PV patients (Table 3). Based on the important parameters obtained from the multivariate analysis, we performed a Kaplan–Meier survival analysis of the effect of TET2 on TFS in ET patients and determined that TET2-mutated ET patients had poorer TFS rates (*p* = 0.008) (Figure 4).

### 3.4. TET2 Mutation in MPN Patients

Among the 125 patients, TET2 mutations were detected in 15 patients (9 ET and 6 PV), with a mutation rate of 12%. Among them, 11 patients combined with the JAK2V617F mutation (6 ET and 5 PV), 2 ET patients combined with the CALR mutation, and 1 ET patient combined with the ASXL1 mutation. Two loci mutations of TET2 were detected in one patient with PV. The median age of patients with TET2 mutations was 53.5 (33–70) years (eight males and seven females). The details of the 15 MPN patients with TET2 mutations are presented in Table 4. The specific sites and complications of thrombotic events in MPN patients with the TET2 mutation are shown in Appendix A.

### 3.5. Comparison of Clinical Characteristics, Coagulation Function and Post-Treatment Thrombosis in TET2-Mutated and -Unmutated ET Patients

Our study showed that TET2 mutation is an independent risk factor for thrombosis in ET patients, and to understand their clinical characteristics and coagulation status, we compared the clinical characteristics and coagulation functions of TET2-mutated and -unmutated patients. We observed that the age of ET patients with TET2 mutations was significantly older than that of patients without TET2 mutations (*p* = 0.031) (Table 5). In terms of coagulation function, TET2-mutation patients had significantly higher D-Dimer (*p* = 0.007), significantly lower antithrombin III (AT-III) (*p* = 0.031), and a higher percentage of D-Dimer (*p* = 0.002) and fibrin degradation products (FDP) (*p* = 0.014) above the normal range compared to ET patients without TET2 mutations (Table 5). All 75 ET patients received antiplatelet or cytoreductive therapy, and the incidence of thrombosis after follow-up treatment was significantly higher in ET patients with TET2 mutations than in those without TET2 mutations (*p* = 0.032) (Table 5).

## 4. Discussion

MPN is a clonal malignant hematologic disease characterized by one or more peripheral blood cell increases as the main clinical feature [1], and the incidence rates of PV, ET and PMF in the US Surveillance, Epidemiology, and End Results database are 10.9/1, 9.6/1, and 3.1/1 million, respectively, with median-age-of-onset values of approximately 65, 68 and 70 years, respectively [19], and median survival rates of approximately 15, 18 and 4.4 years, respectively [20]. Thrombosis is one of the most common complications of MPN, with thrombotic events reported to occur in approximately 30% of patients with PV and 29% of patients with ET [7]. MPN thrombosis can be caused by inflammatory factors that damage the vascular endothelium, increasing blood viscosity due to an increased number of red blood cells, leukocytosis, platelet activation, neutrophil activation, and increased circulating microparticles [21,22,23,24].

In this study, the JAK2V617F mutation rate in ET patients was 64%, which is generally consistent with the results reported by the Mayo Clinic and Song [25,26]. The JAK2 mutation rate in PV patients was 84%, while some studies have reported JAK2 mutation rates ranging from approximately 93 to 98% in PV patients in the United States and European regions [11,25,26]. The exclusion of some patients due to missing information during the pre-screening phase may have contributed to this discrepancy, and this bias was exacerbated by the small-sample-size limitation of this study. In addition, some studies have reported JAK2 mutation rates ranging from about 83 to 90% in patients with PV in Asia [27,28,29], and the results of this study are largely consistent with them; therefore, this difference may also be related to ethnicity and region. In the present study, TET2 and ASXL1 were the most common non-driver mutations, in agreement with most previous results [25,26,30].

The risk factors for thromboembolism in patients with MPN vary by subtype. In ET, age > 60 years, cardiovascular risk factors, history of previous thrombosis and positive JAK2V617F mutations have been identified as risk factors for thrombosis [9]. In addition, many seminal studies have shown an association between other factors and the risk of ET thrombosis, including abnormal karyotype (excluding -Y), neutrophil and monocyte counts, chronic kidney disease, hyperuricemia, and LDL ≥ 70 mg/dL [31,32,33,34,35]. In a cohort of 183 ET patients from the Mayo Clinic, TET2 mutation showed a significant association with ET thrombosis independent of age and driver mutation status, but this association was not evident in either PV or a cohort of 174 ET patients from the University of Florence, Italy [25]. Our study, likewise, demonstrated that TET2 remained an independent risk factor for thrombosis in ET patients with age, cardiovascular risk factors, and previous thrombosis as covariates, again an association not observed in PV. In PV, the multivariate analysis showed that previous arterial events, hyperlipidemia and hypertension predicted subsequent arterial thrombotic events; previous venous events, leukocyte counts ≥ 11 × 10^9^/L, and major bleeding were predictors of subsequent venous events; and TET2 or ASXL1 mutations had no effect on arterial or venous thrombosis [36]. However, in a PV case–control study conducted by Segura-Díaz et al. [13], TET2 mutation was significantly associated with thrombotic events, and patients with TET2 mutations had a significantly higher risk of thrombosis. A recent study of 1508 patients with PV reported that NLR was significantly higher in patients with venous thrombosis, and in a multivariate analysis, NLR values ≥ 5 and previous venous thrombotic events were independently associated with the risk of venous thrombosis in PV patients [37]. Our study also showed that previous thrombotic history and NLR were independent risk factors for thrombosis in PV patients, and that TET2 or ASXL1 did not affect thrombosis in PV patients, even in the univariate analysis. The relationship between non-driver mutations and thrombotic events in MPN is controversial and may be related to the number of cases and regional differences. Our study provides data on thrombotic risk factors in MPN patients in Asia and a reference for future larger studies.

A further analysis of the clinical characteristics of TET2-mutated versus -unmutated ET patients revealed the older age of ET patients with TET2 mutations, consistent with the results obtained by Tefferi et al. [38] and Brousseau et al. [39]. The higher risk of thrombosis in elderly ET patients [9] may explain why ET patients with TET2 mutations are more likely to experience thrombotic events. However, our multivariable analysis using age and other factors as covariates confirmed the predictive value of TET2 on the risk of ET thrombosis, which was not affected by age. In addition, ET patients with TET2 mutations had significantly higher D-Dimers, significantly lower AT-III, and a higher percentage of D-Dimers and FDPs above the normal range, which may explain why TET2 mutation is a risk factor for thrombosis in ET patients; however, a larger sample size is needed to demonstrate this association. It has been shown that the level of P-selectin exposure on platelets of TET2-deficient mice increases in response to stimulation with high concentrations of thrombin [40]. P-selectin, which is usually released when platelets are activated, acts as an adhesion factor on the surface of activated platelets and the vascular endothelium, promoting platelet aggregation and contributing to a hypercoagulable state of blood, and then promotes thrombosis [41]. It has also been shown that TET2-mutation-associated clonal hematopoiesis accelerates the progression of atherosclerosis in mice [42], which is the pathological basis of thrombosis, indirectly reflecting the correlation between TET2 mutation and thrombosis.

However, there are some limitations to our study. This study was retrospective and had a small sample size that did not strictly differentiate between arterial and venous thrombotic events. Nevertheless, we hope that our results will encourage further studies on more patients and provide some basis for future research.

In conclusion, our study reported the risk factors for thrombosis in MPN patients in Asia, and complemented the data on the relationship between non-driver mutations and thrombotic events in Asian MPN patients, providing a reference for future, larger studies. Our data suggest that TET2 mutation is an independent risk factor for thrombosis in ET patients, but no such association is observed in PV. Therefore, TET2 mutation may be more valuable in predicting thrombosis in ET patients compared to PV. ET patients with TET2 mutations were older and had differences in coagulation compared to TET2-unmutated patients. However, prospective studies with more patients are needed to validate our results and to fully elucidate the association.

## Figures and Tables

**Figure 1 jcm-11-06615-f001:**
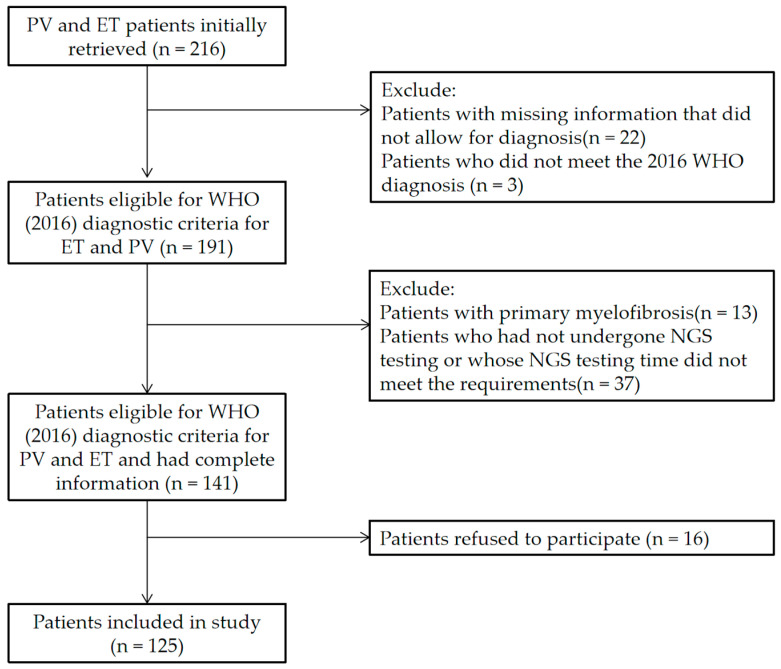
Patient selection flowchart.

**Figure 2 jcm-11-06615-f002:**
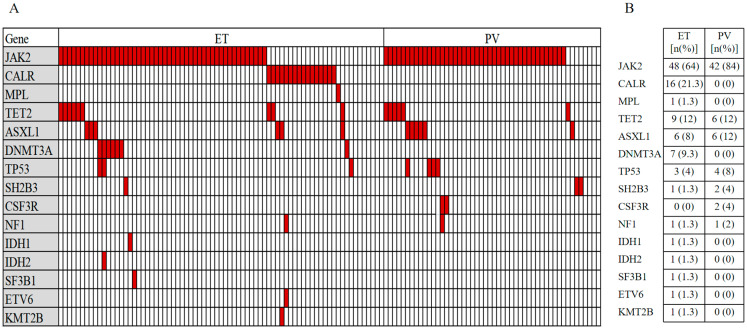
Distribution and frequency of mutated genes in patients with ET (n = 75) and PV (n = 50). (**A**) Co-segregation plot for individual mutations. Each column represents one patient. (**B**) Number and frequency of mutations.

**Figure 3 jcm-11-06615-f003:**
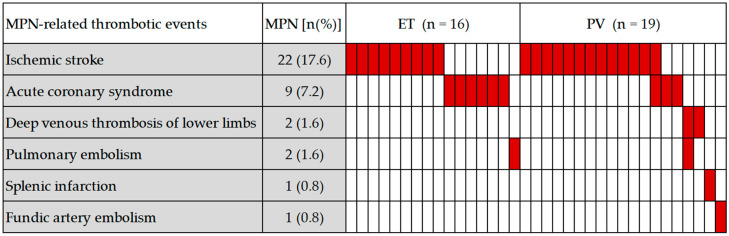
Distribution and frequency of MPN-related thrombotic events. Each column represents one patient.

**Figure 4 jcm-11-06615-f004:**
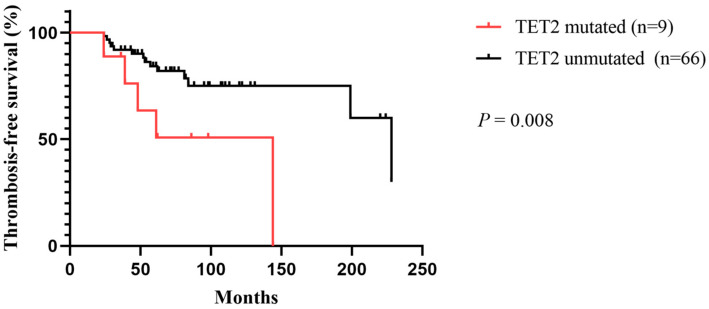
Comparison of TFS in ET patients with and without TET2 mutation.

**Table 1 jcm-11-06615-t001:** Clinical characteristics of 125 patients with MPN.

Clinical Features	Total (n = 125)	ET (n = 75)	PV (n = 50)	*p*
Gender [n (%)]				<0.001
Male	67 (53.6)	29 (38.7)	38 (76.0)	
Female	58 (46.4)	46 (61.3)	12 (24.0)	
Age at dx [year, M(range)]	50 (18–78)	50 (19–76)	50.5 (18–78)	0.393
Age > 60 years [n (%)]	31 (24.8)	15 (20.0)	16 (32.0)	0.246
Blood count at dx				
WBC [×10^9^/L, M(range)]	8.24 (2.95–34.23)	8.15 (2.95–34.23)	8.81(4.45–20.95)	0.301
Leucocytosis [n (%)] *	20 (16.0)	15 (20.0)	5 (10.0)	0.127
HGB [g/L, M(range)]	161 (102–255)	142 (102–169)	190 (163–255)	<0.001
HCT [%, M(range)]	47.3 (31.1–76)	41.6 (31.1–50.2)	54.6 (48.2–76)	<0.001
PLT [×10^9^/L, M(range)]	530 (111–1639)	615 (460–1639)	298.5 (111–819)	<0.001
NLR [M(range)]	2.98 (0.72–15.01)	2.77 (0.72–12.35)	3.54 (1.23–15.01)	0.08
CV risk factor [n (%)] ^#^	63 (50.4)	34 (45.3)	29 (58.0)	0.165
Splenomegaly [n (%)]	42 (33.6)	18 (24.0)	24 (48.0)	0.005
MF grading [n (%)]				
MF-0	96 (76.8)	58 (77.3)	38 (76.0)	0.863
MF-1	24 (19.2)	14 (18.7)	10 (20.0)	0.853
MF-2	5 (4.0)	3(4.0)	2(4.0)	1.000
Driver gene mutation [n (%)]				
JAK2V617F mutation	88 (70.4)	48 (64.0)	40 (80.0)	0.055
JAK2 exon12 mutation	0 (0.0)	0 (0.0)	2 (4.0)	-
CALR mutation	16 (12.8)	16 (21.3)	0 (0.0)	-
MPL mutation	1 (0.80)	1 (1.3)	0 (0.0)	-
Triple negative	18 (14.4)	10 (13.3)	8 (16.0)	0.677
JAK2V617F mutation burden [%, M(range)]	32.25 (3.27–93.28)	22.22 (3.27–75.24)	62.77 (7.5–93.28)	<0.001
Non-driver mutation [n (%)]				
TET2 mutation	15 (12.0)	9 (12.0)	6 (12.0)	1.00
ASXL1 mutation	12 (9.6)	6 (8.0)	6 (12.0)	0.543
Treatments [n (%)]				
Antiplatelet agents	92 (73.6)	48 (64.0)	44 (88.0)	0.003
Cytoreductive treatment	112 (89.6)	64 (85.3)	48 (96.0)	0.056

*p*: Comparison of ET patients and PV patients. * ET: WBC ≥ 11 × 10^9^/L; PV: WBC > 15 × 10^9^/L. ^#^ Patients who had at least one cardiovascular risk factor. ET: essential thrombocythemia; PV: polycythemia vera; Dx: diagnosis; WBC: white blood cell count; HGB: hemoglobin; HCT: hematocrit; PLT: platelet count; NLR: neutrophil-to-lymphocyte ratio; CV: cardiovascular; MF: bone marrow fibrosis.

**Table 2 jcm-11-06615-t002:** Univariate and multivariate analysis of risk factors for thrombosis in MPN patients.

Variable	Total (n = 125)			
	Univariable		Multivariable	
	HR (95% CI)	*p*	HR (95% CI)	*p*
Gender		0.906		
Age at dx	1.05 (1.03–1.08)	<0.001	1.03 (1.00–1.06)	0.046
Age > 60 years at dx	2.99 (1.52–5.88)	0.001		0.282
JAK2V617F mutation	2.63 (1.02–6.79)	0.045		0.476
CALR mutation		0.17		
Triple negative		0.704		
TET2 mutation	3.03 (1.42–6.49)	0.004	2.84 (1.26–6.36)	0.011
ASXL1 mutation		0.203		
Antiplatelet agents		0.062		
Cytoreductive treatment		0.155		
Leucocytosis *^†^		0.207		
WBC ^†^	1.07 (1.01–1.13)	0.019		0.682
HGB ^†^		0.078		
HCT ^†^		0.225		
PLT ^†^		0.717		
NLR ^†^	1.16 (1.06–1.28)	0.002	1.18 (1.06–1.32)	0.003
CV risk factor ^#^	2.12 (1.04–4.34)	0.039		0.376
History of remote thrombosis	4.59 (1.85–11.39)	0.001	5.33 (1.93–14.70)	0.001
MF-1		0.98		
Splenomegaly		0.914		

* ET: WBC ≥ 11 × 10^9^/L; PV: WBC > 15 × 10^9^/L. ^†^ Blood cell count at diagnosis. ^#^ Patients who had at least one cardiovascular risk factor. HR: hazard ratio; CI: confidence interval; Dx: diagnosis; WBC: white blood cell count; HGB: hemoglobin; HCT: hematocrit; PLT: platelet count; NLR: neutrophil-to-lymphocyte ratio; CV: cardiovascular; MF: bone marrow fibrosis.

**Table 3 jcm-11-06615-t003:** Univariate and multivariate analysis of risk factors for thrombosis in ET and PV patients.

Variable	ET (n = 75)				PV (n = 50)			
	Univariable		Multivariable		Univariable		Multivariable	
	HR (95% CI)	*p*	HR (95% CI)	*p*	HR (95% CI)	*p*	HR (95% CI)	*p*
Gender		0.927				0.152		
Age at dx	1.06 (1.01–1.10)	0.012		0.071		0.069		
Age > 60 years at dx	3.54 (1.21–10.35)	0.021		0.136		0.209		
JAK2V617F mutation		0.144				0.189		
CALR mutation		0.231						
Triple negative		0.754				0.299		
TET2 mutation	3.89 (1.35–11.27)	0.012	4.10 (1.40–12.01)	0.01		0.164		
ASXL1 mutation		0.497				0.268		
Antiplatelet agents		0.095				0.925		
Cytoreductive treatment		0.271				0.512		
Leucocytosis *^†^		0.116				0.502		
WBC ^†^		0.101				0.096		
HGB ^†^		0.914				0.792		
HCT ^†^		0.651				0.636		
PLT ^†^		0.846				0.46		
NLR ^†^		0.059			1.13 (1.00–1.27)	0.046	4.77 (1.33–17.16)	0.017
CV risk factor ^#^	3.14 (1.00–9.81)	0.049		0.179		0.45		
History of remote thrombosis	6.19 (1.35–28.46)	0.019	6.89 (1.45–32.68)	0.015	3.48 (1.08–11.2)	0.037	1.67 (1.03–1.32)	0.014
MF-1		0.946				0.97		
Splenomegaly		0.581				0.87		

* ET: WBC ≥ 11 × 10^9^/L; PV: WBC > 15 × 10^9^/L. ^†^ Blood cell count at diagnosis. ^#^ Patients who had at least one cardiovascular risk factor. HR: hazard ratio; CI: confidence interval; Dx: diagnosis; WBC: white blood cell count; HGB: hemoglobin; HCT: hematocrit; PLT: platelet count; NLR: neutrophil-to-lymphocyte ratio; CV: cardiovascular; MF: bone marrow fibrosis.

**Table 4 jcm-11-06615-t004:** TET2 mutation information in 15 MPN patients.

No.	Exon	VAF (%)	Nucleotide Change	pHGVS	Mutation Type	Disease	Gender	Age * (Year)	Other Gene Mutation	Thrombosis
1	6	1.71	c.3629T>C	p.L1210P	Nonsynonymous	ET	Female	57	JAK2V617F	Acute coronary syndrome
2	6	43.99	c.3782G>A	p.R1261H	Nonsynonymous	ET	Female	45	JAK2V617F	Ischemic stroke
3	3	2.14	c.1602_1605del	p.R534fs	Frameshift	ET	Female	37	JAK2V617F	None
4	10	18.49	c.4250T>G	p.V1417G	Nonsynonymous	ET	Female	50	JAK2V617F	Pulmonary embolism
5	9	47.33	c.4145A>C	p.H1382P	Nonsynonymous	ET	Male	67	JAK2V617F	Acute coronary syndrome
6	3	1.05	c.3344delC	p.P1115fs	Frameshift	ET	Male	66	JAK2V617F	Acute coronary syndrome
7	3	37.14	c.540_541insATTC	p.Q180fs	Frameshift	ET	Female	69	CALR	None
8	3	48.29	c.86C>G	p.P29R	Nonsynonymous	ET	Male	50	CALR	None
9	3	1.47	c.1860C>A	p.Y620X	Nonsynonymous	ET	Female	70	ASXL1	None
10	6	2.3	c.3781C>T	p.R1261C	Nonsynonymous	PV	Male	61	JAK2V617F	Ischemic stroke
11	5	36.28	c.3508C>T	p.Q1170X	Nonsynonymous	PV	Male	70	JAK2V617F	None
12	3	36.7	c.822delC	p.I274fs	Frameshift	PV	Male	49	JAK2V617F	Ischemic stroke
13	3	1.11	c.3010delA	p.K1005Rfs *2	Frameshift	PV	Male	38	-	Ischemic stroke
14	3	24.5	c.2158C>T	p.Q720 *	Nonsynonymous	PV	Male	33	JAK2V617F	None
15	4	1.39	c.3431A>C	p.E1144A	Nonsynonymous	PV	Female	66	JAK2V617F	Ischemic stroke
	5	42.49	c.3541_3544del	p.I1181fs	Frameshift					

* Age at diagnosis. VAF: variant allele frequency.

**Table 5 jcm-11-06615-t005:** Clinical features and coagulation function of TET2-mutated and -unmutated ET patients.

Clinical Features	TET2 Mutated (n = 9)	TET2 Unmutated (n = 66)	*p*
Gender [n (%)]			1.000
Male	3 (33.3)	26 (39.4)	
Female	6 (66.7)	40 (60.6)	
Age at dx [year, M(range)]	57 (37–70)	48.5 (19–76)	0.031
Age > 60 years [n (%)]	4 (44.4)	11 (16.7)	0.072
WBC [×10^9^/L, M(range)]	8.54 (5.28–34.23)	7.97 (2.95–19.99)	0.254
WBC > 11 × 10^9^/L [n (%)]	3 (33.3)	12 (18.5)	0.375
HGB [g/L, M(range)]	142 (102–165)	142 (104–169)	0.895
HCT [%, M(range)]	42.2 (31.1–48.7)	41.6 (32.9–50.2)	0.627
PLT [×10^9^/L, M(range)]	641 (460–993)	614 (462–1639)	0.552
NLR [M(range)]	3.64 (1.96–8.17)	2.71 (0.72–12.35)	0.203
CV risk factor [n (%)] *	6 (66.7)	28 (42.4)	0.285
Splenomegaly [n (%)]	3 (33.3)	15 (77.3)	0.441
MF-1 [n (%)]	2 (22.2)	12 (18.5)	0.672
JAK2V617F mutation	6 (66.7)	42 (63.6)	1.000
CALR mutation	2 (22.2)	14 (21.2)	1.000
Coagulation function index ^#^			
D-Dimer [mg/L, 0–0.55, M(range)]	0.46 (0.22–2.13)	0.26 (0.08–0.85)	0.007
PT [s, 10–14, M(range)]	12.1 (11.2–13.8)	11.7 (10.4–13.7)	0.26
PTA [%, 80–120, M(range)]	95.4 (80–105.7)	99.7 (80.8–116.7)	0.113
INR [0.8–1.15, M(range)]	1.05 (0.97–1.21)	1.01 (0.9–1.2)	0.112
APTT [s, 23.3–32.5, M(range)]	30.5 (25–37.6)	30.2 (24.7–46.3)	0.756
TT [s, 13–21, M(range)]	18.4 (15.3–20.3)	18.55 (14.9–21.8)	0.585
Fbg [g/L, 1.7–4.05, M(range)]	2.24 (1.86–4.06)	2.62 (1.33–4.62)	0.529
FDP [mg/mL, 0–5, M(range)]	2 (0.7–7.24)	2 (0.6–2.72)	0.519
AT-III [%, 75–125, M(range)]	89 (76.1–124.6)	99.75 (81.2–116.7)	0.031
D-Dimer > 0.55 mg/L [n (%)]	4 (44.4)	2 (3.1)	0.002
PT > 14s [n (%)]	0 (0.0)	0 (0.0)	-
PTA > 120% [n (%)]	0 (0.0)	0 (0.0)	-
INR > 1.15 [n (%)]	1 (11.1)	8 (12.5)	1.000
APTT > 32.5 s [n (%)]	3 (33.3)	15 (23.4)	0.680
TT > 21 s [n (%)]	0	5 (7.8)	1.000
Fbg > 4.05 g/L [n (%)]	1 (1.11)	2 (3.1)	0.330
FDP > 5 mg/mL [n (%)]	2 (22.2)	0 (0.0)	0.014
AT-III > 125% [n (%)]	0 (0.0)	0 (0.0)	-
Thrombosis after receiving treatment [n (%)]	4 (44.4)	8 (12.1)	0.032

* Patients who had at least one cardiovascular risk factor. ^#^ TET2-mutated ‘N’ evaluable = 9; TET2-unmutated ‘N’ evaluable = 64.Dx: diagnosis; WBC: white blood cell count; HGB: hemoglobin; HCT: hematocrit; PLT: platelet count; NLR: neutrophil-to-lymphocyte ratio; CV: cardiovascular; MF: bone marrow fibrosis; PT: prothrombin time; PTA: prothrombin time activity percentage; INR: international normalized ratio; APTT: activated partial thromboplastin time; TT: thrombin time; Fbg: fibrinogen; FDP: fibrin degradation products; AT-III: antithrombin III.

## Data Availability

The datasets generated during and/or analyzed during the current study are available from the corresponding author on reasonable request.

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
