# Peer review of "TET2 Mutation May Be More Valuable in Predicting Thrombosis in ET Patients Compared to PV Patients: A Preliminary Report"

_jcm, 2022, doi:10.3390/jcm11226615_

Round 1

Reviewer 1 Report

Comments to the authors,

Wang and colleagues are sharing their data on MPN (ET and PV) cases tested for non-driver mutations by NGS analyses. They tried to correlate their findings with thrombotic events. They found that TET2 mutations were the most common additional mutations and were correlated to the risk of thrombosis in ET patients, only. This paper is interesting. Please find enclosed my questions and comments.

- In the Method section, there is no indication about the timing of thromboses that are analysed in the study (before or after diagnosis). In the same way, there is no indication about the timing of NGS analyses too (at or after diagnosis, before or after thrombotic event…).

- As 125 patients have been analysed in this study, and due to the reassessment of diagnosis for this study (see Method section) please provide a flow-chart of global numbers of cases to see the exhaustivity of cases and the risk of biases for case selection.

- In Table 1, please check the mutational numbers, 125, 128, 53, 50 patients… Also, check the non-driver mutational numbers particularly for ASXL1 cases.

- Please explain once again why the percentage of JAK2-negative PV cases is so elevated. In West countries, those cases are very rare (less than 2%) not 16%... As diagnoses have been made according to WHO classification, please provide additional data on PV criteria for those negative cases. Some cases could really be secondary PG or constitutional PG…

- In the same way, in Table 1, among ET cases, some patients have Hb or Ht levels at 16.9 and 50.2% respectively, please provide information about misdiagnosed PV or masked PV among them.

- In Table 1, why only MF1 graduation can be seen? What about MF0 or 2?

- The splenomegaly rates are impressive, where those abnormalities seen by palpation or US scan?

- There is no information about low and high-risk classification of cases, no information about cytoreductive or antithrombotic drug uses. Are those patients with TET2/thrombotic cases all low-risk cases or high-risk cases with no cytoreduction? You cannot talk about thrombosis without talking about treatment, important biases of interpretation.

- Table 3 is cut in the PDF version.

- As TET2 mutation are correlated to thrombosis, as observed that no recurrent mutation is observed in this serie, please provide some more explanations about the common complication represented by thrombosis.

- When were realized the coagulation analyses? As a complete paragraph and some lines in a Table 5 have been highlighted here, please provide a specific paragraph in the Method section.

- In Table 5, please provide more details about normal range of both D-dimer and AT-III dosages. Give informations about how many cases were defined as above the normal ranges. Significance of median values and then comparisons will not be the same if in the normal range or above. Due to the provided numbers, authors cannot really conclude about hypercoagulable state of TET2 cases…

- In the Discussion section, authors confirmed the differences between their findings and those already published on the same topic. Please add some more sentences about explanations of such differences and the pertinence to publish their data (not only a question of racial description as provided in the Discussion section).

Author Response

Thank you for your comments on our contributions.  We have revised our document according to your comments.  Please refer to the attachment. 

Reviewer 2 Report

This a retrospective single-centre report of incidence and associations of thrombotic events in 125 MPN patients, including 75 with ET and 50 with PV. The main conclusions drawn are that TET2 mutations increase the risk of thrombosis in ET, but not in PV. I think the main issue is that these associations can only really be speculative in a study of this size - here we have only 15 patients with TET2 mutations, and a total of 9 thrombotic events between them, so hoping to differentiate between ET and PV within those numbers is understandably challenging. The findings do contribute to the wider field however, allowing for the limitations imposed by small patient numbers, the single centre retrospective nature of the study and by purely South Asian ethnicity. 

More specific points to be addressed by the authors:
1) The incidence of JAK2 mutations of only 80% in the 'PV' patients is surprisingly low. There should be more explanation of how the diagnosis of PV was made in these 'JAK2 negative cases'

2) The lengths of follow-up are unclear. This should be more clearly indicated. Can thrombotic events can be expressed as events/yr of follow up in different populations?

3) In 50 PV patients you have 40 with JAK2 and 13 who are 'triple negative' - these figures don't add up? Please check and amend

4) I'm not clear whether the incidence of thrombotic events in MPN patients refers to the sum of 'MPN-related' and 'remote' events of just the more recent MPN-related ones - This should be made clear

5) When referring to the N:L ratio as a prognostic marker does this refer to the N:L ratio at the time of the thrombotic event or at some other timepoint? This should be made clear. Same applies to Hb, platelet count etc. 

6) Table 3: The right hand columns are missing from the pdf I accessed for this review so I couldn't view the whole content. This needs to be addressed. 

Author Response

Thanks for your comments on our manuscript. We have revised our paper according to your comments. Please see the attachment.

Reviewer 3 Report

In the current study, Wang et al. retrospectively explore risk factors for thrombotic events in 125 myeloproliferative neoplasm (MPN) patients, including 75 essential thrombocythemia (ET) cases and 50 polycythaemia vera (PV) cases. The results of multivariable analyses show that the non-driver mutation TET2 and a medical history of remote thrombosis are predictive for a higher rate of  thrombotic events in ET patients, whereas the neutrophil-to-lymphocyte ratio (NLR) and the remote thrombosis history are the independent risk factors for thrombosis in PV patients. The authors also report that ET patients with mutant TET2 show higher age and D-Dimer levels as well as lower levels of antithrombin III as compared to those with wild-type TET2, further indicating the role of TET2 mutation in predicting thrombotic events in ET patients. Although this research is of clinical interest and suggests the potential thrombotic risk factors in MPN patients, I have several major concerns regarding applicability of the presented data:

Major comments:

1.       In the multivariate analyses, the authors include the factors age, MPN-related driver mutations, TET2 mutation, ASXL1 mutation, blood parameters, NLR, cardiovascular risk factor, history of remote thrombosis, bone marrow fibrosis and splenomegaly. These included factors are already relatively comprehensive for multivariable analyses. However, there is no information about cytogenetics. As explained in the background, abnormal karyotypes (excluding -Y) have been correlated with higher risk of ET thrombosis. Therefore, it would make this story more complete if the standard diagnostic parameter cytogenetics could be includes into the analyses.

2.       Due to the limited patient cohort size, there are only 15 patients harboring TET2 mutations among 125 MPN patients, including 9 cases with ET and 6 cases with PV. Therefore, the current cohort size is not really strong enough to support the conclusion that TET2 mutations are more valuable in predicting thrombosis in ET patients than in PV. Therefore, it would be better if the authors could enlarge the cohort size.

3.       In this study, the authors mainly analyzed the association between the risk factors of thrombosis and the occurrence frequency of thrombotic events. It would be interesting if the authors could explore the relationship between thrombosis-free survival (TFS) and these risk factors. For example, the research from Cerquozzi et al. (DOI 10.1038/s41408-017-0035-6) indicates that prior thrombotic events and leukocytosis were predictive of subsequent thrombotic events in general, whereas non-driver mutations (TET2 and ASXL1) were not significantly associated with neither arterial nor venous thrombotic events in PV patients. Therefore, the current research could be further improved if the authors could specifically analyze the association between TFS and thrombotic risk factors, especially for TET2 mutation.

4.       For MPN patients with thrombosis, the mainstay of management consists on preventing hemostatic complications, by antiplatelet and/or anticoagulant treatment and myelosuppressive agents in high-risk patients. In this case, it would further improve this study if the response of patients to antiplatelet and/or anticoagulant treatment could be summarized and compared between the cohorts with and without thrombotic risk factors.

Minor comments:

1.       In the section “Incidence of thrombotic events in MPN patients”, the authors describe the frequency of various thrombotic events in ET and PV patients, respectively. It would be better and more visualized if the authors could summarize these data into a figure.

2.       The manuscript definitely has to be revised by an English native speaker.

Author Response

(The authors gave the same response as above.)

Round 2

Reviewer 1 Report

All the comments and questions have been answered.

Author Response

Thanks again for your advice and guidance!

Reviewer 2 Report

I thank the authors for addressing the comments that I made in my initial review (and those of the other reviewers).

My main reservation about this paper remains that the robustness of the relationships between TET2 mutation status and thrombotic risk may be being overstated (especially the differential importance between ET and PV) given that the sample size is so small. The wording of the conclusion should be toned down to make these reservations clearer, for example the sentence: 'Therefore, TET2 mutation is more valuable in predicting thrombosis in ET patients compared with PV'  - it is too early to state this so explicitly on the basis of these preliminary data. 

Author Response

Response: As Reviewer suggested that we have revised this statement.

Reviewer 3 Report

Although the authors were not able to enlarge the study cohort, some of my other comments were addressed.

Author Response

(The authors gave the same response as above.)
